



# Complex rift patterns, a result of interacting crustal and mantle weaknesses, or multiphase rifting? Insights from analogue models

Frank Zwaan[1], Pauline Chenin[2], Duncan Erratt[2], Gianreto Manatschal[2], Guido Schreurs[1]

[1]Institute of Geological Sciences, University of Bern, Baltzerstrasse 1+3, 3012 Bern, Switzerland
[2]Université de Strasbourg, CNRS, ITES, UMR 7063, 5 rue Descartes, Strasbourg F-67084, France

*Correspondence to*: Frank Zwaan (frank.zwaan@geo.unibe.ch, fzwaan@hotmail.com)

**Abstract.** During lithospheric extension, localization of deformation often occurs along structural weaknesses inherited from previous tectonic phases. Such weaknesses may occur in both the crust and mantle, but the combined effects of these weaknesses on rift evolution remains poorly understood. Here we present a series of 3D brittle-viscous analogue models to test the interaction between differently oriented weaknesses located in the brittle upper crust and/or upper mantle. We find that crustal weaknesses usually express first at the surface with the formation of graben parallel to their orientation; then, structures parallel to the mantle weakness overprint them and often become dominant. Furthermore, the direction of extension exerts minimal control on rift trends when inherited weaknesses are present, which implies that present-day rift orientations are not always indicative of past extension directions. We also suggest that multiphase extension is not required to explain different structural orientations in natural rift systems. The degree of coupling between the mantle and upper crust affects the relative influence of the crustal and mantle weaknesses: low coupling enhances the influence of crustal weaknesses, whereas high coupling enhances the influence of mantle weaknesses. Such coupling may vary over time due to progressive thinning of the lower crustal layer, as well as due to variations in extension velocity. These findings provide a strong incentive to reassess the tectonic history of various natural examples.

## 1. Introduction

Over the course of continental rifting, initial localization of deformation often occurs along structural weaknesses inherited from previous tectonic phases (e.g. Wilson 1966; Morley et al. 1990; Nelson et al. 1992; Bonini et al. 1997; Corti 2012). These initial weaknesses may be situated anywhere in the lithosphere, although structural heterogeneities tend to be attenuated or erased at great depth where temperature is high (Braun et al 1999; Yamakasi et al. 2006). On Earth, regions of stable continental lithosphere are usually dominated by two strong layers: the brittle upper crust and the brittle upper lithospheric mantle, decoupled from each other by a weaker, ductile lower crust (e.g. Brun 1999; Burov and Watts 2006; Burov 2011; Zwaan et al. 2019). Inherited weaknesses situated in either of these strong layers are most likely to be reactivated and affect subsequent rift architecture (Chenin and Beaumont 2013). As a result, initial deformation may localize



independently in both competent lithospheric layers and ultimately will have to interact as the ductile lower crust progressively thins during extension (e.g. Sutra et al. 2013).

Crustal and mantle weaknesses can have very different orientations and relative weakness. Weaknesses in the upper lithospheric mantle may represent large-scale tectonic features such as major suture zones to microscopic fabric related to

preferred orientations of olivine crystals (e.g. Tommasi et al. 2009). The (upper) continental crust often contains a variety of (often older) tectonic lineaments, thrusts, faulted basins and structural orientations that may have no direct relation to the underlying mantle weaknesses. For instance the various structural orientations occurring in the crust of the European plate near the Alps are mostly unrelated to the Alpine suture zone (e.g. Dèzes et al 2004; Schori et al. *in review* and references therein). Moreover, the opening of an ocean as part of a Wilson cycle does not always fully reuse the original suture zone

from the preceding orogenic event (Wilson 1966; Krabbendam 2001; Chenin et al. 2019), but may choose to follow other lithospheric weaknesses that are more suitable oriented or deform more readily. Similarly, crustal structures are not always reactivated (e.g. Ring 1994; Bell et al. 2014, Claringbould et al. 2017 and references therein). The continental lithosphere may thus contain a variety of structural inheritances and old pre-existing weaknesses that may or may not play a role in subsequent rift tectonics.


Tectonic modelling provides an excellent tool to test how differently oriented weaknesses in the crust and mantle interact and affect rift systems, yet this topic has received only limited attention so far. Some numerical modellers have tested the effects of weaknesses in both the crust and mantle in 2D, showing that the presence, shape and location of lithospheric weaknesses may strongly affect the general style of rifting or allow rifting to take place altogether (Dyksterhuis et al. 2007;

Chenin and Beaumont 2013; Liao and Gerya 2015; Wenker and Beaumont 2018; Chenin et al. 2019). However, these 2D models lack the 3D component associated with different structural orientations. Some other numerical modellers have investigated the impact of linear weaknesses in three dimensions, revealing that deformation will follow an obliquely oriented crustal weakness, but that individual rifts will cross the weakness and local horizontal block rotations may occur within the rift zone (Van Wijk 2005; Brune and Autin et al. 2013; Duclaux et al. 2020). However these 3D numerical

modelling studies did not test the competition and interaction between weaknesses in the crust and mantle.

Analogue modelling studies of (pseudo-)2D and 3D rift processes provide some similar insights to the aforementioned numerical studies, however these modelling efforts also either focused on structural inheritance in the crust or on the effect of mantle weaknesses (e.g. Brun and Tron 1993; Le Calvez and Vendeville 2002, Bellahsen and Daniel 2005; Autin et al.

2010, 2013; Agostini et al. 2011; Zwaan et al. 2016, 2018, 2019; Molnar et al. 2017, 2018, 2019; Wang et al. *in review*). To our knowledge, only Molnar et al. (2020) simulated differentially oriented mantle and crustal weaknesses in their experiments to reproduce the Red Sea rift. Their results suggest that inherited crustal weaknesses may localize transform faults, whereas the overall orientation of the rift is generally controlled by a weak thermal mantle anomaly. However the



authors only tested a limited set of parameters (i.e. they only completed one model with both mantle and crustal weakness
that were 30˚ oblique to each other).

The aim of this study is therefore to determine the relative impact of mantle and crustal structural inheritance on the localization of deformation during early continental rifting (e ≤ 10% or β ≤ 1.1) by means of analogue tectonic models at the Tectonic Modelling Laboratory of the University of Bern. We used time-lapse photography, Particle Image Velocitmetry
(PIV) and X-Ray CT scanning methods for model analysis, allowing a thorough insight into model evolution. The main result of our modelling efforts is that crustal and mantle weaknesses can simultaneously localize rift structures leading to intricate fault patterns that may otherwise be interpreted as a result of multiphase extension. Therefore, we conclude that multiphase extension is not required to explain different structural orientations in rift basins, which provides a strong incentive to reassess the tectonic history of a number of natural examples.


## 2. Methods

### 2.1. Materials

We used both brittle and viscous materials to simulate the brittle and ductile parts of a 30 km thick crust belonging to a stable continental lithosphere. A 3 cm thick layer of fine quartz sand (ø = 60-250 μm) represented a 22.5 km brittle upper
crust (following a 3:1 upper crust-to-lower crust thickness ratio expected in a stable lithosphere, e.g. Zwaan et al. 2019). The sand was sieved from ca. 30 cm height into the experimental apparatus, ensuring a constant brittle layer density of ca. 1560 $kg/m^3$. We also flattened the sand using a scraper at every cm during the build-up of the model in order to avoid lateral variations in layer thickness during model preparation.

We applied a 1 cm thick viscous layer to simulate a 7.5 km thick ductile lower crust below the brittle upper crust. This layer consisted of a mixture of near-Newtonian ($\eta$ = ca. $1.5 \cdot 10^5$ Pa·s; n = 1.05-1.10, Zwaan et al. 2018c) SGM-36 Polydimethyl-siloxane (PDMS) and corundum sand ($\rho_{specific}$ = 3950 $kg/m^3$, Carlo AG 2019).  Both components were mixed according to a 0.965 : 1.00 weight ratio. The density of this viscous mixture was similar to the density of the overlying sand layer (ca. 1600 $kg/m^3$), so that the model density profile was analogue to the density profile expected in nature, preventing the unrealistic
buoyancy effects that pure, low-density PDMS ($\rho_{specific}$ = ca. 960 $kg/m^3$) can cause. Further material properties are presented in Table 1.



**Table 1. Model materials**

| Granular materials | Quartz sand[a] | Corundum sand[b] |
|---|---|---|
| Grain size range (ø) | 60-250 μm | 88-125 μm |
| Specific density ($\rho_{specific}$)[c] | 2650 kg/m$^3$ | 3950 kg/m$^3$ |
| Sieved density ($\rho_{sieved}$) | 1560 kg/m$^3$ | 1890 kg/m$^3$ |
| Angle of internal peak friction ($\phi_{peak}$) | 36.1˚ | 37˚ |
| Coefficient of internal peak friction ($\mu_{peak}$)[d] | 0.73 | 0.75 |
| Angle of dynamic-stable friction ($\phi_{dyn}$) | 31.4˚ | 32˚ |
| Coefficient of dynamic-stable friction ($\mu_{dyn}$)[d] | 0.66 | 0.62 |
| Angle of reactivation friction ($\phi_{react}$) | 33.5˚ | - |
| Coefficient of reactivation friction ($\mu_{react}$)[d] | 0.66 | - |
| Cohesion (C) | 9 ± 98 Pa | 39 ± 10 Pa |
| **Viscous materials** | **Pure PDMS[a, e]** | **PDMS/corundum sand mixture[a]** |
| Weight ratio PDMS : corundum sand | - | 0.965 kg : 1.00 kg |
| Density (ρ) | 965 kg/m$^3$ | ca. 1600 kg/m$^3$ |
| Viscosity (η) | ca. 2.8·10$^4$ Pa·s | ca. 1.5·10$^5$ Pa·s[f] |
| Type[f] | Newtonian (n = ca. 1)[g] | near-Newtonian  (n = 1.05-1.10)[g] |

[a]    Quartz sand, PDMS and viscous mixture characteristics after Zwaan et al. (2016; 2018b, c)
[b]    Corundum sand characteristics after Panien et al. (2006)
[c]    Specific densities after Carlo AG (2020)
[d]    $\mu = \tan(\phi)$
[e]    Pure PDMS rheology details after Rudolf et al. (2016)
[f]    Viscosity value holds for model strain rates < 10$^{-4}$ s$^{-1}$
[g]    Power-law exponent n (dimensionless) represents sensitivity to strain rate





**2.2. Set-up**

The basic set-up for our experiments involved a mobile base plate underlying the model materials (Fig. 1a, b). The edge of the base plate created a so-called "velocity discontinuity" (VD) as soon as the sidewall to which it was attached moved outward orthogonally away from the model axis. This VD has often been used to simulate a (linear) discontinuity in the strong upper lithospheric mantle in magma-poor rift systems (e.g. Allemand and Brun 1991; Tron and Brun 1991; Brun and

Tron 1993; Michon and Merle 2000; Gabrielsen et al. 2016; Zwaan et al. 2019). We applied two VD orientations, either parallel, or 30˚ oblique to the model axis for model series 1 and 2, respectively (the VD's orientation being defined by angle $\theta_{VD}$, Fig. 1c, d, Table 1).

Linear weaknesses in the sand layer representing the upper crust were induced by means of either pre-cut faults or "seeds".

Pre-cut faults were simply fault planes cut in the sand (e.g. Bellahsen and Daniel 2005; Tong et al. 2014), in our case by means of fish wire. The fish wire with a diameter of 0.25 mm was placed on top of the viscous layer and after model construction was completed, removing the wire resulted in a c. 1 mm wide zone of perturbed sand grains. The initial dip of the pre-cut faults was 60˚ (Andersonian normal fault). Along these fault planes, the coefficient of reactivation friction was lower than the coefficient of internal peak friction (0.73 vs. 0.66, Table 1). Hence the fault plane was ca. 9.3% weaker than

the surrounding sand layer, thus localizing deformation more easily. Seeds were small semi-circular ridges of viscous material with a diameter of 0.5 cm, placed on top of the viscous layer representing the lower crust (e.g. Le Calvez and Vendeville 2002; Zwaan and Schreurs 2017, Molnar et al. 2019, 2020). Above these seeds the standard sand layer was 33% thinner and thus 44% weaker, localizing deformation (Zwaan et al. 2020). Similar to the VD, these crustal weaknesses were oriented in different directions, either parallel ($\theta_{CW} = 0°$) or oblique ($\theta_{CW} = 30°$ or -30°) to the model axis, but instead of a

single weakness, we applied a sequence of parallel weaknesses (Fig. 1e, f, Table 1). Note that in the case of a model axis-parallel VD, a 30˚ oblique crustal weakness orientation would be the mirror image of a -30˚ orientated seed, thus representing the same set-up that was not necessary to repeat. Next to the models with crustal weaknesses, we also present results of models without such weaknesses for reference purposes.

As extension is applied by moving the sidewall outward, a regional orthogonal extension field develops. This extension field could be oblique to both the simulated mantle and crustal weaknesses, as a function of their respective orientation (Fig. 1c-e, Table 1). The standard extension velocity is a constant 2 cm/h. Also the effects of faster extension (v = 4 cm/h) was tested (Sub-series 2.2, see Table 1), however the final bulk extension was always 3 cm (so that e = 10% or β = 1.1 at the end of a model run, given an initial model width of 30 cm). Similarly, some models explored the effects of a thicker viscous layer of

2.5 cm, but the total model height was always kept at 4 cm (i.e. the sand layer was 0.5 cm thinner in these models). A total of 15 models are included in this paper, some of which were re-runs of selected models in a CT-scanner (Table 1). Repeating these selected models also allowed an assessment of model reproducibility.



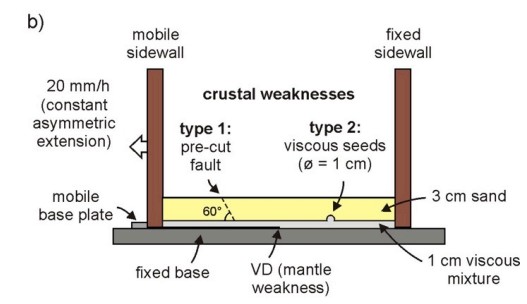

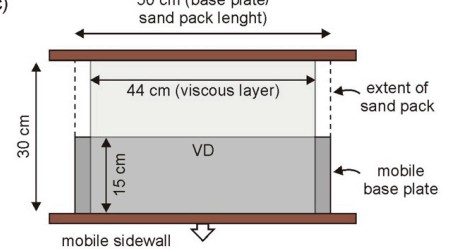

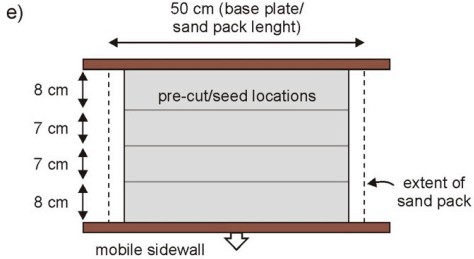

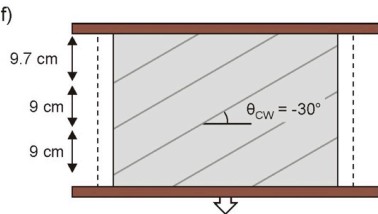


**Fig. 1. Model set-up. (a) 3D sketch of general set-up. VD: velocity discontinuity representing a weakness or discontinuity in the strong upper mantle. (b) Section view depicting standard model layering and the two types of crustal weaknesses (pre-cuts and viscous seeds). (c-d) Model set-up) and base plate geometries shown in map view. (c) Base plate configuration for series 1 (with VD parallel to model axis or $\theta_{VD} = 0°$). (d) Base plate configuration for series 2 (with VD 30° oblique to model axis or $\theta_{VD} = 30°$). (e-f)**
**Examples of crustal weakness geometries at the top of the viscous layer, shown in map view. (e) Model axis-parallel crustal weaknesses ($\theta_{CW} = 0°$). (f) Crustal weaknesses -30° oblique to the model axis ($\theta_{CW} = -30°$).**



**Table 2. Model parameters**

| | | Model | VD orientation (angle $\theta_{VD}$)* | Simulated crustal weaknesses | | | Layer thickness | | Extension velocity | Results shown in |
|---|---|---|---|---|---|---|---|---|---|---|
| | | | | Orientation (angle $\theta_{CW}$)* | Type | | Brittle | Viscous | | |
| Series 1 (orthogonal VD) | Series 1.1 | A | 0° | - | - | | 3 cm | 1 cm | 2 cm/h | Figs. 2, 11 |
| | | B | 0° | -30° | Pre-cut** | | 3 cm | 1 cm | 2 cm/h | Figs. 2, 11 |
| | | C | 0° | -30° | Seed | | 3 cm | 1 cm | 2 cm/h | Figs. 2, 11 |
| | | D$^{CT}$ | 0° | -30° | Seed | | 3 cm | 1 cm | 2 cm/h | Figs. 2, 3 |
| | Series 1.2 | E | 0° | - | - | | 2.5 cm | 1.5 cm | 2 cm/h | Figs. 4, 11 |
| | | F | 0° | 0° | Seed | | 2.5 cm | 1.5 cm | 2 cm/h | Figs. 4, 11 |
| Series 2 (30° oblique VD) | Series 2.1 | G | 30° | - | - | | 3 cm | 1 cm | 2 cm/h | Figs. 5, 11 |
| | | H$^{CT}$ | 30° | - | - | | 3 cm | 1 cm | 2 cm/h | Figs. 5, 6 |
| | | I | 30° | 0° | Pre-cut | | 3 cm | 1 cm | 2 cm/h | Figs. 5, 11 |
| | | J | 30° | 0° | Seed | | 3 cm | 1 cm | 2 cm/h | Figs. 5, 11 |
| | | K | 30° | -30° | Seed | | 3 cm | 1 cm | 2 cm/h | Figs. 7, 11 |
| | | L$^{CT, \#}$ | 30° | -30° | Seed | | 3 cm | 1 cm | 2 cm/h | Figs. 7, 8 |
| | | M | 30° | 30° | Seed | | 3 cm | 1 cm | 2 cm/h | Figs. 7, 11 |
| | Series 2.2 | N | 30° | 0° | Seed | | 3 cm | 1 cm | 4 cm/h | Figs. 9, 11 |
| | | O$^{CT, \#}$ | 30° | 0° | Seed | | 3 cm | 1 cm | 4 cm/h | Figs. 9, 10 |


| | |
|---|---|
| * | VD: velocity discontinuity, representing a mantle weakness, CW: (simulated) crustal weakness |
| ** | Pre-cut fault inclination 40° instead of 60° |
| CT | CT-scanned models |
| # | No 3D images available due to technical issues during the model run |




### 2.3. Model monitoring and analysis

Basic monitoring of model evolution for all models was done by means of time lapse photography. We applied a Nikon D200 (10 MPx) camera for map view images, and two obliquely oriented Nikon D810 (36.3 MPx) cameras providing a

stereoscopic view of the model after every 1/3 mm of extension (i.e. after every min, or in the case of Models N and O with double extension velocity, after every 30 s). A 4 x 4 cm grid of thin < 1 mm thick corundum sand applied on the model surface allowed for visual assessment of horizontal displacements. Note that Models L and O lack the oblique photographs due to technical issues, so that only map view pictures were available (Table 2).

These photographs not only provided a visual impression of surface model evolution, but also allowed more detailed analysis and quantification of deformation by means of Particle Image Velocitmetry (PIV) techniques (e.g. Adam et al., 2005, Boutelier et al. 2019 and references therein).The PIV analysis was performed using DaVis 10.2 software from LaVision, which compares the time lapse images to derive vector fields of horizontal displacements over time. These horizontal displacement vector fields were subsequently applied to extract incremental maximum normal strain, which is taken as a

proxy of active deformation in the model at specific time-steps. We applied the high-resolution Nikon D810 oblique photographs for PIV analysis (after correction for the deformation due to the obliquity). Since no oblique photographs were available for models L and O, we used the lower quality Nikon D200 photographs for PIV analysis instead.

In addition to the PIV analysis, we re-ran four selected models a CT scanner (Table 1) to obtain insights into their internal

structural evolution. CT-scanning exploits the differences in attenuation (mostly a function of density) allowing the visualization of features inside an otherwise opaque object (e.g. Naylor et al. 1994; Colletta et al 1991; Schreurs et al. 2003). The scanning was done using the Siemens Somaton Definition AS X-Ray CT scanner of the University of Bern Institute of Forensic Medicine. We scanned the models at intervals of 2.5 mm of extension, providing a complete picture of their evolution. The CT data were subsequently visualized and analysed by means of open-source software (Horos

https://horosproject.org/).





### 2.4. Scaling

Model scaling procedures serve to make sure that the experiment properly represents the natural prototype.Brittle materials have time-independent rheology, so that the main concern for scaling purposes is the angle of internal friction of the sand (36.1˚), which is similar in models and nature (31-38˚, Byerlee 1978, Table 3). Yet when scaling viscous materials, one needs to take into account their time-dependent rheology. We start with the basic formula addressing the stress ratios between model and nature ($\sigma_{model}$/ $\sigma_{nature}$): $\sigma^* = \rho^* \cdot h^* \cdot g^*$, where $\rho^*$, $h^*$ and $g^*$ are density, length and gravity ratios,

respectively (Hubbert 1937; Ramberg 1981). We can then acquire the strain rate ratio $\dot{\varepsilon}^*$ from the stress and viscosity ratios $\sigma^*$ and $\eta^*$ (Weijermars and Schmeling 1986): $\dot{\varepsilon}^* = \sigma^*/\eta^*$. Knowing the strain rate ratio, the following equations yield the velocity and time ratios ($v^*$ and $t^*$): $\dot{\varepsilon}^* = v^*/h^* = 1/t^*$. When assuming a relatively high lower crustal viscosity of ca. $5 \cdot 10^{21}$ Pa·s that may be representative for early rift systems, e.g. Buck 1991), one hour in our models thus translates to ca. 3 Myr in nature and our standard model velocity (20 mm/h) scales up to ca. 5 mm/y. These values are similar to typical rift extension

velocities (several mm/y, e.g. Saria et al. 2014, Fig. 1b). An overview of scaling parameters is provided in Table 3.

    An additional scaling test concerns the dynamic similarity of the model with respect to nature. Dynamic similarity between the brittle model layer and its upper crustal equivalent is estimated with the ratio $R_s$ between the gravitational stress and cohesive strength or cohesion C (Ramberg 1981; Mulugeta 1998): $R_s$ = gravitational stress/cohesive strength = $(\rho \cdot g \cdot h) / C$. A

cohesion of 12 MPa for natural materials in the upper crust, combined with a 9 Pa cohesion in the sand, yields a $R_s$ of 51 for both model and nature. This natural cohesion value of 12 MPa is slightly lower than cohesions measured in rock deformation labs (e.g. Handin, 1969; Jaeger and Cook 1976; Twiss and Moore 1992), but is acceptable, since in nature, the lithosphere is generally weakened by several phases of deformation. When assessing viscous materials, the Ramberg number $R_m$ applies (Weijermars and Schmeling 1986): $R_m$ = gravitational stress/viscous strength = $(\rho \cdot g \cdot h^2) / (\eta \cdot v)$. The Ramberg number for

both the model and natural materials is 51. With both our Rs and Rm model values very similar to those in nature, we consider our models adequately scaled for the simulation of continental rift tectonics.





**Table 3. Scaling parameters**


| | | Model | Nature |
|---|---|---|---|
| **General parameters** | Gravitational acceleration (g) | 9.81 m/s$^2$ | 9.81 m/s$^2$ |
| | Extension velocity (v) | 5.6·10$^{-5}$ m/s | 1.6·10$^{-10}$ m/s |
| **Brittle layer** | Material | Quartz sand | Upper crust |
| | Peak internal friction angle ($\varphi_{peak}$) | 36.1˚ | 30-38˚ |
| | Thickness (h) | 3·10$^{-2}$ m | 2.25·10$^4$ m |
| | Density ($\rho$) | 1560 kg/m$^3$ | 2800 kg/m$^3$ |
| | Cohesion (C) | 9 Pa | 10$^7$ Pa |
| **Viscous/ ductile layer** | Material | PDMS/corundum sand mix | Lower crust |
| | Thickness (h) | 1·10$^{-2}$ m | 7.5·10$^4$ m |
| | Density ($\rho$) | 1600 kg/m$^3$ | 2900 kg/m$^3$ |
| | Viscosity ($\eta$) | 1.5·10$^5$ Pa·s | 5·10$^{21}$ Pa·s |
| **Dynamic scaling values** | Brittle stress ratio ($R_s$) | 51 | 51 |
| | Ramberg number ($R_m$) | 17 | 17 |



## 3. Results

### 3.1. Analysis of Series 1.1 experiments: impact of the nature of crustal weaknesses

The PIV-derived incremental strain evolution of models A-D from series 1.1 (with model axis-parallel VD) shows that all

models develop rift structures, but clear differences occurred as a function of the presence and type of simulated crustal weaknesses (Fig. 2). Reference Model A, without simulated crustal weaknesses, developed early on two deformation zones on both sides of VD that became clearly visible on PIV images after 5 mm of extension (e = 1.7%, Fig. 2a). These deformation zones subsequently localized along discrete faults bounding a set of two parallel graben on each side of the VD after 10 mm of extension (e = 3.3%, Fig. 2b). This system remained in place until the end of the model run (30 mm of

extension, β = 10%), when a new normal fault started to appear away from the model axis (Fig. 2d).

Introducing a series of oblique pre-cut faults in Model B caused important changes in our model results, segmenting and increasing the distance between the VD-parallel graben and inducing a set of secondary oblique graben structures. The -30˚ oriented pre-cut faults were reactivated early on (i.e. after ca. 5 mm of extension or e = 1.7%) and two additional

deformation zones parallel to the VD developed (similar to those observed in Model A, Fig 2a, although some discrete faults already appeared within these zones at this early stage, Fig. 2f). As the model developed and normal faulting further localized along the VD-parallel deformation zones, the interference of the pre-cut faults with the VD-parallel normal faults caused the latter to appear to be slightly offset (Fig. 2g-j). This resulted in a segmentation of the graben along the model axis, which were straight in reference Model A (Fig. 2a-e). It also induced a greater distance between both VD-parallel graben

compared to the reference model (ca. 4 cm vs. 0 cm, Fig. 2d, e, i, j). In addition, minor secondary (half-)graben structures developed along the pre-cut seeds (Fig. 2i, j).

Applying -30˚ oriented seeds in Models C and D (both having the same set-up) created a very similar segmentation and graben offset to Model B (where crustal weaknesses are pre-cut faults), although this segmentation was expressed more

visibly in Models C and D. At an early stage (5 mm of extension or e = 1.7%), the VD-parallel deformation zones developed, but strain clearly localizes along the seeds as well, creating small graben at an angle to that generated by the VD (Fig. 2k, p). Furthermore, where both systems interfered, strain along the seed-parallel graben faults was enhanced (Fig. 2k, p). As extension continued, two VD-parallel graben formed in the centre of the model, but these graben were significantly segmented and appear to be offset by the well-developed oblique graben structures parallel to the seeds (Fig. 2l-o, q-t).


The CT scans of Model D provide additional insights into the model's internal evolution (Fig. 3). As shown by the PIV results, the CT data clearly indicated that the seeds localize early faulting after ca. 5 mm of extension (e = 1.7%, Fig. 3a-c). The 3D CT imagery also reveals that the VD generated a general rift zone along the axis of the model early on (Fig. 3c), which is not well-defined in section view (Fig. 3a, b), but the borders of which are visible in the shape of VD-parallel



deformation zones on PIV top views (Fig. 2a, f, k, p). This rift zone became more expressed over time as normal (boundary)

faults started to appear, after 10 mm of extension (e = 3.3%, Fig. 3d, e). At the same moment, the two graben parallel to the

VD seen on PIV-derived strain maps (Fig. 2l-n, q-s), are also clearly visible on both sections and 3D images (Fig. 3e, f).

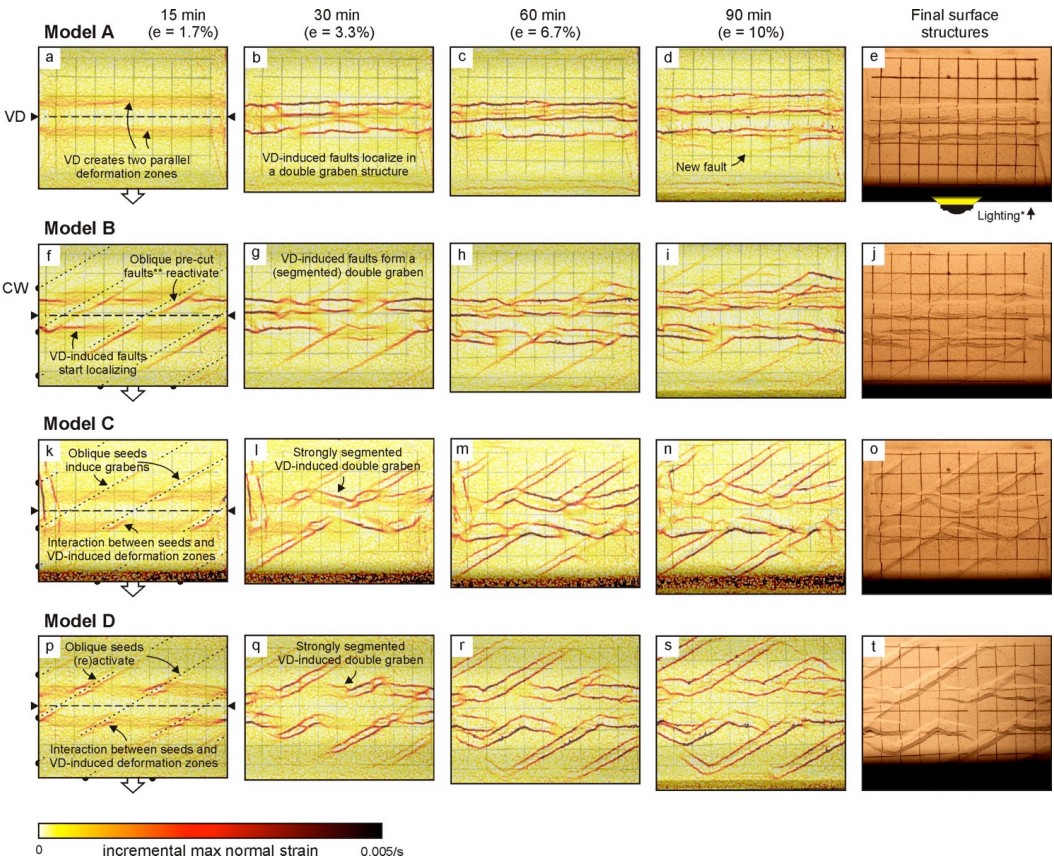


**Fig. 2. Incremental maximum normal strain maps of Series 1.1 models obtained through PIV analysis (with model axis-parallel**
**VD and -30° oblique weaknesses in the brittle cover, if applied) obtained through PIV analysis, taken as a proxy of fault activity.**
**The increments for PIV analysis were 1 min, i.e. every 0.33 mm of extension. The original model map view pictures are visible in**
**the background. The right-hand column depicts the model surface at the end of each experiment. VD: velocity discontinuity: CW:**
**(simulated) crustal weakness, * shadows cast by one-sided lighting allowing a better visualization of the final surface structures, ***
**the locations of the pre-cut faults at the base of the sand layer are indicated; the offset of their surface expression is due to their**
**inclination (Fig. 1b).**





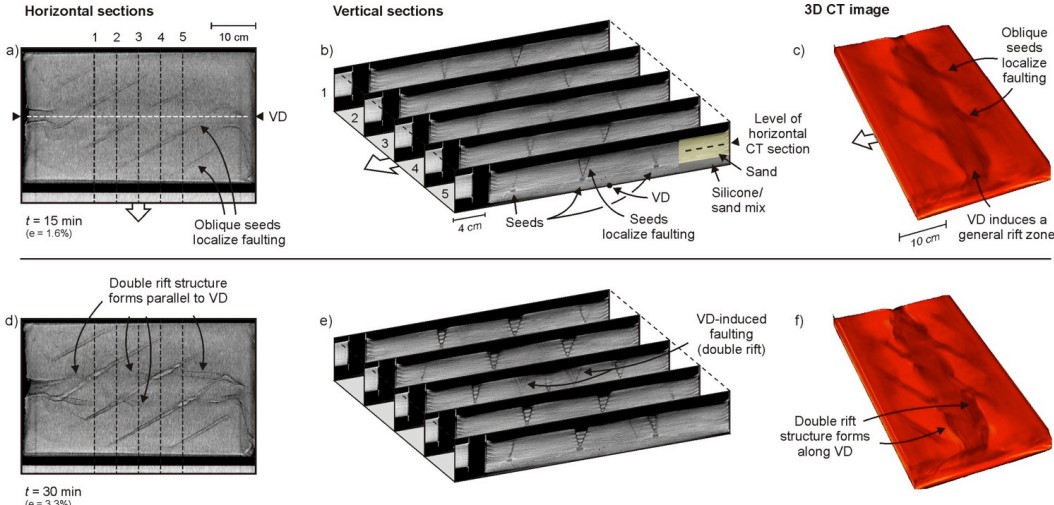


**Fig. 3. CT analysis of Model D (model axis-parallel VD, -30° oblique seeds). Left column: horizontal sections through the brittle layer. Middle column: serial cross-sections. Right column: 3D CT images. Note that the colours in these images represent CT-scanning intensities and are not to be interpreted as a measure of model surface altitude.**

**3.2. Analysis of Series 1.2 experiments: impact of viscous lower crust thickness**

Models E and F from series 1.2 contained a slightly thicker viscous layer than the other models (1.5 cm instead of 1 cm, Table 2), but still provided insights into the effects of seeds oriented parallel to the VD, which was itself aligned to the model axis (Fig. 4). Model E, without seeds or pre-cuts, developed a double graben somewhat similar to Model A, although the graben in Model E are farther apart, flanking a central horst (Figs. 2a-e, 4a-e). When adding the model-axis parallel seeds in

Model F, the outermost seeds localize deformation and the double graben structure (Fig 4f-j) became even wider when compared to Model E (Fig. 4a-e). We furthermore observe the late-stage development of a normal fault structure in the middle of the central horst block of Model F, along the central seed (Fig. 4i, j). Both Models E and F develop some boundary effects at the short ends of the set-up, where the sand talus partially collapses, creating minor graben perpendicular to the main structures (Figs. 2a, 4).






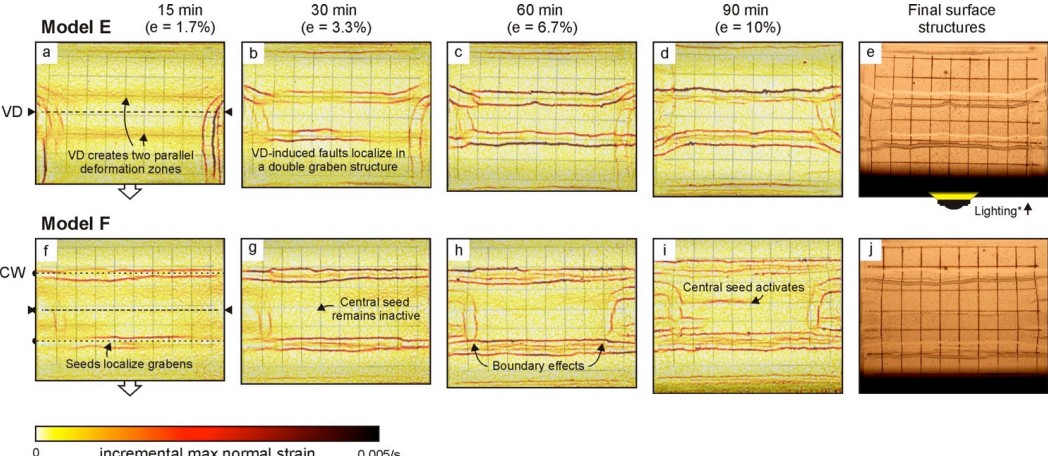

**Fig. 4. Incremental maximum normal strain maps of Series 1.1 models** (with model axis-parallel VD, and equally oriented seeds, if applied) obtained through PIV analysis, taken as a proxy of fault activity. Note that these models have a thicker (1.5 cm) viscous layer and a thinner (2.5 cm) brittle cover than the other models in this paper. The increments for PIV analysis were 1 min, i.e. every 0.33 mm of extension. The original model map view pictures are visible in the background. The right-hand column depicts the model surface at the end of each experiment. The final column depicts the model surface view at the end of each experiment. VD: velocity discontinuity: CW: (simulated) crustal weakness, * shadows cast by one-sided lighting allowing a better visualization of the final surface structures

### 3.3. Analysis of Series 2.1 experiments: impact of VD obliquity with respect to extension

The results from Models G and H, with a 30° oblique VD but without simulated crustal weaknesses, provide a reference for our series 2.1 models (Fig. 5a-j). Both these models produced two deformation zones parallel to the VD in their initial stages (5 mm of extension or e = 1.7%), very similar to the deformation zones observed in Model A (Figs. 2a, 5a, f). However, Models G and H developed a series of right-stepping en echelon normal faults along these initial deformation zones, connected by relay ramps, which contrasts with the through-going normal faults in Model A (Figs. 2a-e, 5a-j). These en-echelon normal faults form the boundary of the rift zone, which continued to widen over time while the faults grew in length (Fig. 5a-j).

The CT-scans of model H provide high quality visualization of both the complex surface and internal structural model evolution resulting from an oblique VD (Fig. 6). We observe the early initiation of the en-echelon faults after 5 mm of extension (e = 1.7%) in vertical CT sections, and the development of the rift zone between these en-echelon faults in 3D view (Fig. 6b-f). In addition, the CT scans reveal that two graben structures developed on both sides of the rift zone,





reminiscent of the double graben system found in Models A and E (both with model-parallel VD but no simulated crustal

weaknesses, Figs. 2a-e, 4a-e).

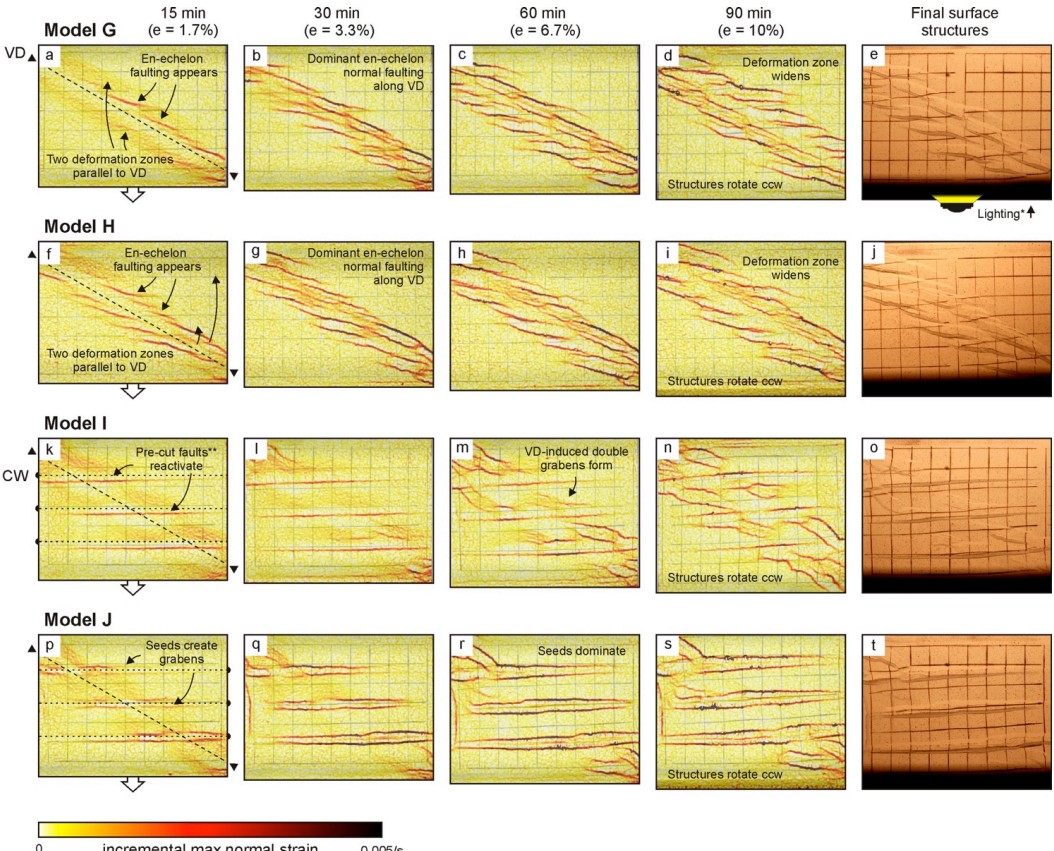

**Fig. 5. Incremental maximum normal strain maps of models G-J from Series 2.1 (with 30° oblique VD and model axis-parallel**
**crustal weaknesses in the brittle cover, if applied) obtained through PIV analysis, taken as a proxy of fault activity. Note that these**
**models have a thicker (1.5 cm) viscous layer and a thinner (2.5 cm) brittle cover than the other models in this paper. The**
**increments for PIV analysis were 1 min, i.e. every 0.33 mm of extension. The original model map view pictures are visible in the**
**background. The right-hand column depicts the model surface at the end of each experiment. VD: velocity discontinuity: CW:**
**(simulated) crustal weakness, ccw: counter clockwise* shadows cast by one-sided lighting allowing a better visualization of the**
**final surface structures, ** the location of the pre-cut faults at the base of the sand layer are indicated, the offset of their surface**
**expression is due to their inclination (Fig. 1b).**


Introducing crustal weaknesses in Models I and J strongly affected the subsequent rift structures by segmenting them and by taking up significant amounts of deformation (Fig. 5k-t), even more so than in series 1.1. We found that the pre-cut faults in

Model I strongly localized deformation, so much so that the reactivated pre-cut faults dominate the model surface structures (Fig. 5k-o). Note that these faults (similarly to Model B, Fig. 2f-j) did not develop symmetric graben but half-graben as only a single boundary fault was present (Fig. 5l-o). Yet, we also observed the early development of VD-parallel deformation zones and eventually of double graben between the dominant pre-cut faults (Fig. 5k-o). Applying seeds as crustal weaknesses localized deformation even more, to the point that almost no features related to the VD were present (Fig. 5p-t).

This contrasts with the observations from our Series 1.1 models, in which the VD-(sub-)parallel graben remained well-developed at all times (Figs. 2-3). Furthermore we observed a general counter-clockwise rotation in map view in Models G-J (Fig. 5d, e, i, j, n, o, s, t).

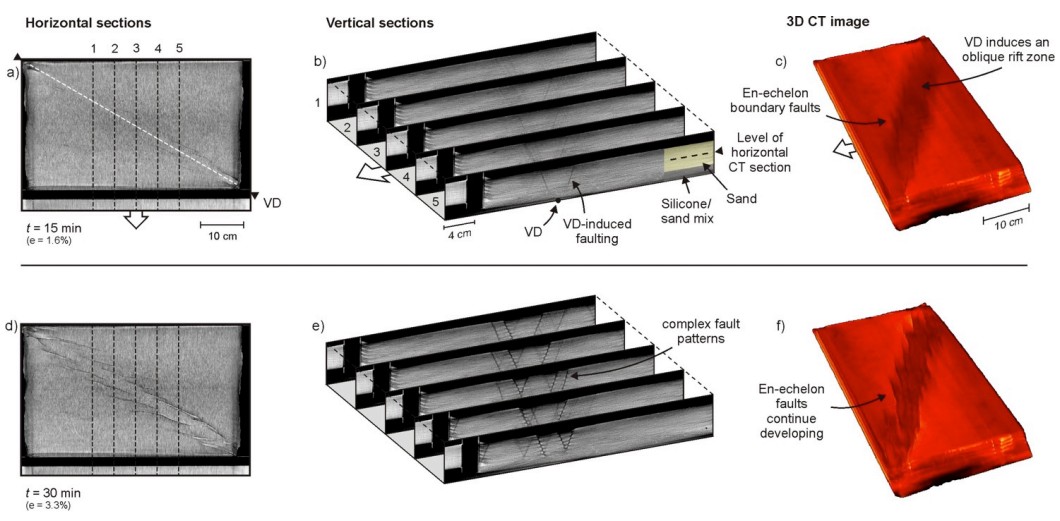


**Fig. 6. CT analysis of Model H (30° oblique VD, no crustal weaknesses simulated). Left column: horizontal sections through the brittle layer. Middle column: serial cross-sections. Right column: 3D CT images. Note that the colours in these images represent CT-scanning intensities and are not to be interpreted as a measure of model surface altitude.**

The results from the additional models with a -30° oblique VD highlight the effect of seed orientation on the final model structure in this type of experiment (Models K, L and M; Figs. 7 and 8). Model K, with a -30° oblique series of seeds developed a VD-parallel deformation zone and a series of right-stepping en-echelon double graben following the VD trend, intersected and apparently offset by a second series of smaller graben along the seeds (Fig. 7a-j). These smaller seed-induced graben appeared later on the strain maps than the graben forming above the model axis-parallel seeds in Model J (Fig. 5p-t),





and the initial stage of Model K was rather similar to Models G and H (30˚ oblique VD but without simulated crustal weaknesses, Fig. 5a, f, 7a). As the deformed grid of the final surface images revealed, the normal faults in Model K had a moderate oblique slip component (Fig. 7e). Model L had the same set-up as Model K, but no high-resolution photographs were available for similarly detailed PIV analysis. However the PIV results (and final surface structures) match the general features described for Model K (Fig 7a-j).


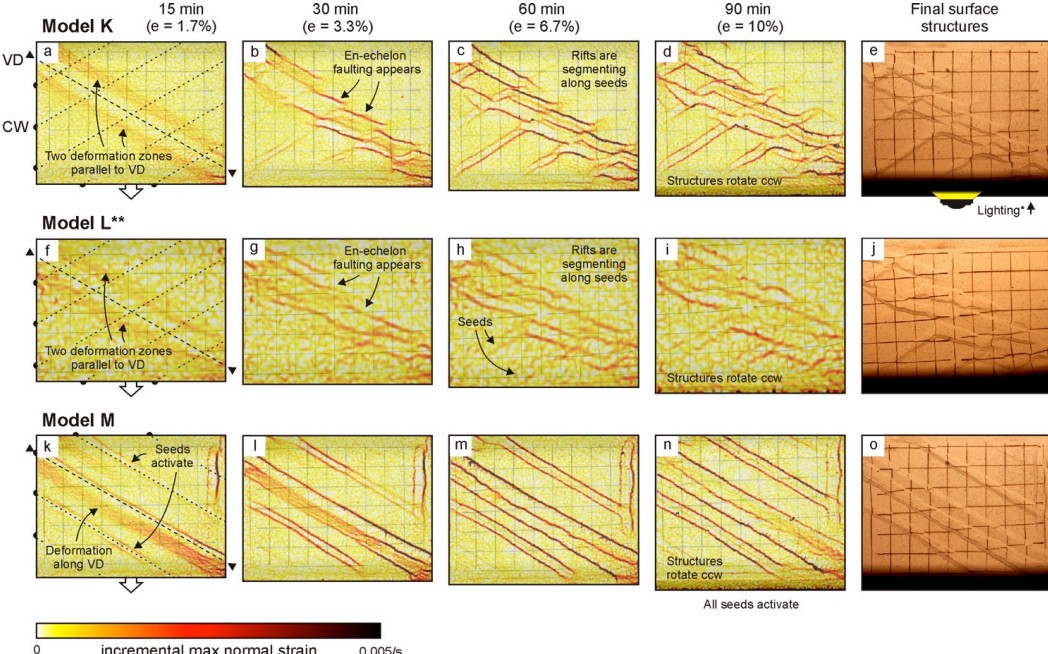

**Fig. 7. Incremental maximum normal strain maps of models K-M from Series 2.1 (with 30˚ oblique VD and oblique seeds in the brittle cover) obtained through PIV analysis, taken as a proxy of fault activity. Note that these models have a thicker (1.5 cm)**
**viscous layer and a thinner (2.5 cm) brittle cover than the other models in this paper. The increments for PIV analysis were 1 min, i.e. every 0.33 mm of extension. The original model map view pictures are visible in the background. The right-hand column depicts the model surface at the end of each experiment. VD: velocity discontinuity: CW: (simulated) crustal weakness, ccw: counter clockwise, * shadows cast by one-sided lighting allowing a better visualization of the final surface structures, ** due to technical issues, the PIV analysis of Model L is less detailed.**


Despite the lack of PIV resolution, the CT images of Model L revealed the early evolution of this segmentation and the associated complex internal structure (Fig. 8). We found that even if the PIV analysis did not show clear faulting along the seeds after 5 mm of extension (e = 1.7%), the CT sections reveal that the seeds already started localizing deformation (Fig.



8a, b), but not to a sufficient degree to appear on the strain maps (Fig. 7a, f). As the model run continued, the CT data show

the evolution of increasingly complex (interacting) fault systems that became challenging to visualize by vertical CT sections, but are reasonably distinct on horizontal CT sections and well defined in 3D CT images (Fig. 8d-f).

Finally, Model M showed how a set-up in which both the VD and seeds are 30˚ oblique, highly linear graben developed instead of the en-echelon basins observed in Models G and H without seeds (Figs. 5a-j, 7k-o). In contrast to the previous

models with an oblique VD, Model M did not develop a double deformation zone parallel to the VD during the initial stage of deformation; instead only a single deformation zone appeared after 5 mm of extension (e = 1.7%, Fig. 7k). As the model continued deforming, this deformation zone evolved into a large graben structure, flanked by additional graben induced by the seeds that already localized in the earlier phase of the model run (Fig. 7k-o). As with Models K and L, the deformed surface grid indicated a minor oblique slip component (Fig. 7e, j, o).


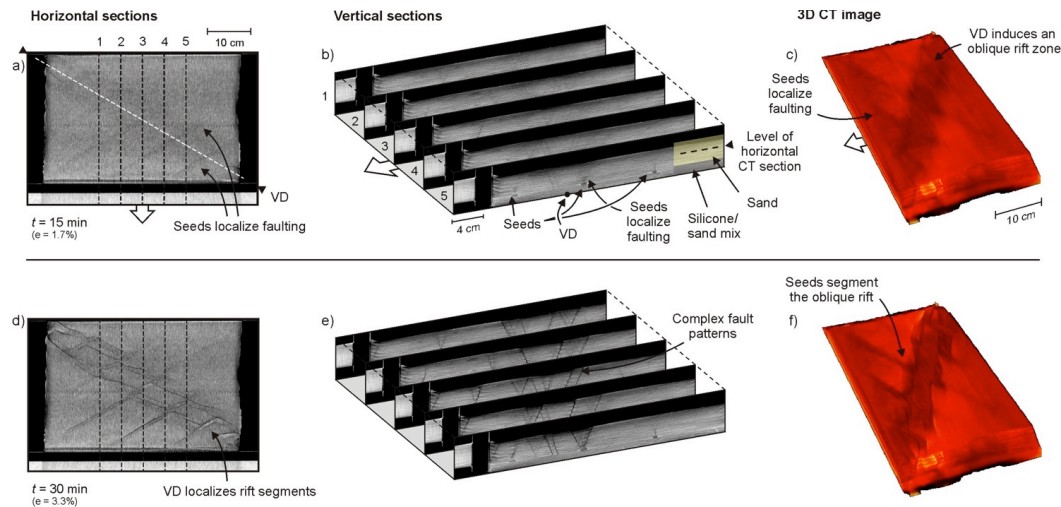

**Fig. 8. CT analysis of Model L (30˚ oblique VD, -30˚ oblique seeds). Left column: horizontal sections through the brittle layer. Middle column: serial cross-sections. Right column: 3D CT images. Note that the colours in these images represent CT-scanning**
**intensities and are not to be interpreted as a measure of model surface altitude.**



### 3.4. Analysis of Series 2.2 experiments: impact of the extension velocity

Besides the orientation of simulated mantle and crustal weaknesses, the extension velocity is an important factor affecting the influence of the VD on subsequent rift evolution, as shown by Series 2.2 Models N and O (Fig. 9). In both these models we use the same set-up as Model J, with the difference that the extension velocity in Models N and O was twice as high as in

Model J (40 mm/h instead of the standard 20 mm/h, Figs. 5p-t, 9). As a result, we found that the seed-induced graben became less dominant in favour of the VD-parallel deformation zones and the subsequent double graben structures along the VD (Figs. 5p-t, 9). The VD-related structures were even better developed than in Model I, which had only pre-cut faults (Figs. 5k-o, 9). As the high-velocity models evolved, a general counter clockwise rotation occurred (Fig. 9d, i). Although the resolution of the PIV analysis results from Model O is reduced due to technical issues, they support the general impression

obtained from Model N (Fig. 9).

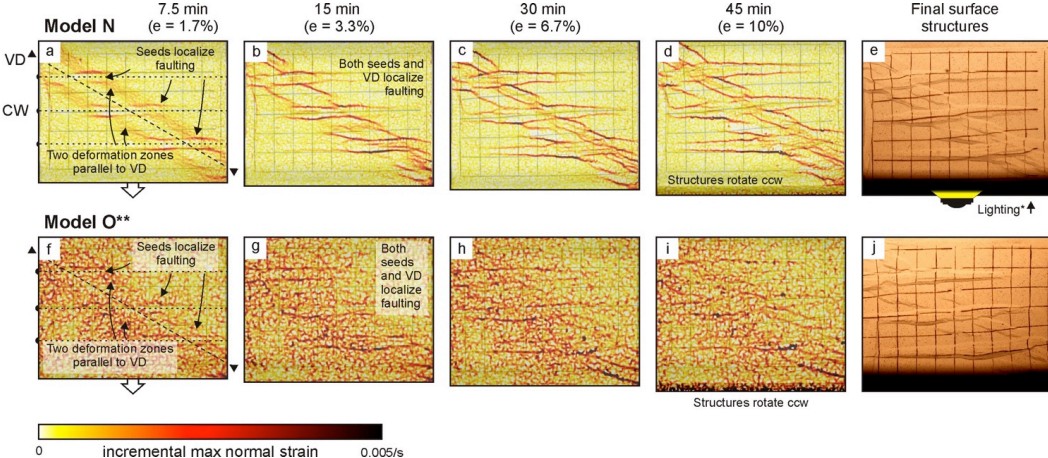

**Fig. 9. Incremental maximum normal strain maps of models N and O from Series 2.2 (with 30° oblique VD, model axis-parallel**
**weaknesses in the brittle cover and a double extension velocity of 40 mm/h, compare to reference models G and H, Fig. 5a-j)**
**obtained through PIV analysis, taken as a proxy of fault activity. The increments for PIV analysis were 1 min, i.e. every 0.33 mm**
**of extension. The original models map view pictures are visible in the background. The right-hand column depicts the model**
**surface at the end of each experiment. VD: velocity discontinuity: CW: (simulated) crustal weakness, ccw: counter-clockwise, ***
**shadows cast by one-sided lighting allowing a better visualization of the final surface structures, ** due to technical issues, the PIV**
**analysis of Model O is less detailed.**





The CT imagery of Model O reveals its early internal evolution associated with the complex interaction between the VD-induced and seed-controlled rift structures (Fig. 10). Similar to the PIV results, CT imagery also shows that the early evolution of Model O was characterized by discrete faulting above the seeds after 5 mm of extension (e = 1.7%), together with more distributed deformation along a central rift zone parallel to the VD (Figs. 9, 10a-c). As deformation continued, deformation along the VD also localized faulting (Fig. 10d-f). After 20 mm of extension (e = 6.7%), a clear series of double graben was visible on both sides of the VD, interrupted by the seed-parallel graben (Fig. 10g-i).


**Fig. 10. CT analysis of Model O (30° oblique VD, model axis-parallel seeds, doubled extension velocity). Left column: horizontal sections through the brittle layer. Middle column: serial cross-sections. Right column: 3D CT images. Note that the colours in these images represent scanning intensities and are not to be interpreted as a measure of model surface altitude.**






## 4. Discussion

### 4.1. Summary of model results

The results of our brittle-viscous analogue models illustrate the effects of differently oriented mantle and crustal weaknesses
on rift structures, as well as the effects of extension velocity (Figure 11). Firstly, the general orientation of the VD (mantle weakness), when no seeds or pre-cuts were present, had a clear impact on subsequent rift structures (Fig. 11a-c). We found that when the VD was aligned with the model axis and oriented orthogonally to the general extension direction, it initially induced two deformation zones parallel to the VD (Fig. 11a, b). These deformation zones subsequently localized normal faulting in a double graben structure, and the horizontal distance between these graben were larger in models with a thicker
viscous layer (Fig. 11b). An obliquely oriented VD also created two initial deformation zones parallel to the VD (Fig. 11c), but in contrast to models with a VD parallel to the model axis, an oblique VD induced a subsequent series of right-stepping en echelon graben, which partially aligned to be sub-perpendicular to the direction of regional extension (Fig. 11 a-c). All oblique VD models also registered a slight counter-clockwise rotation along the VD.

Adding different types of structures that simulate inherited crustal weaknesses to the brittle layer had an important effect on rift development. The model results illustrate that the pre-cut faults, which were characterized by a weakening of the sand layer by ca. 9.3% (difference in friction coefficient, Section 2.2 and Table 1), partially overprinted the general deformation zone and subsequent double graben pattern associated with the VD (Fig. 11d, e). This overprinting was even more pronounced in the models with viscous seeds, which represented a weakening of the sand layer by 44% (Zwaan et al. 2020),
therefore more strongly localizing deformation and faulting in the brittle layer (Fig. 11f-j).

The orientation of the simulated weaknesses with respect to both the regional extension direction and other weaknesses had an important effect on their subsequent (re-)activation (Fig. 11 d–k). Both the VD and crustal structures had the strongest influence when they were oriented orthogonally to the direction of regional extension, the ideal setting for normal fault
development (Fig. 11d-j). When either the VD or simulated crustal weaknesses were oriented obliquely to the regional extension direction, they had less control on the resulting rift. Yet when the VD and the simulated crustal weaknesses were parallel to each other, both effectively localize deformation, even when they were oriented obliquely to the direction of regional extension (Fig. 11j). However, the final structures of these models with VD-parallel crustal weaknesses were very different from the structures in the reference models with no crustal weaknesses at all; while en-echelon faults sub-
orthogonal to the extension direction eventually form in models with only a VD, no graben develop sub-orthogonal to the extension direction when both crustal weaknesses and mantle VD are parallel and at angle with the extension direction (compare Fig. 11c and Fig. 11j).





**Effects of mantle weaknesses only**

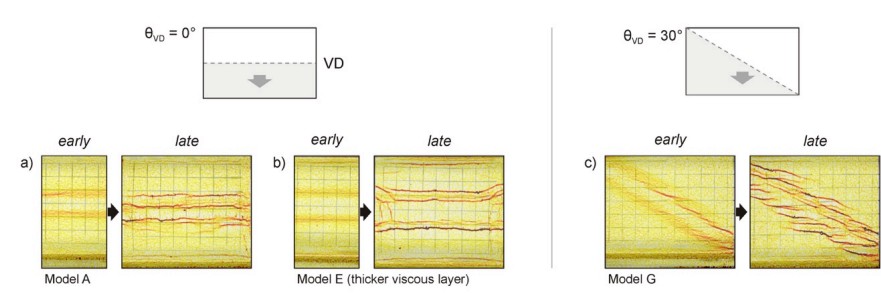

**Combined effects of mantle and crustal weaknesses**

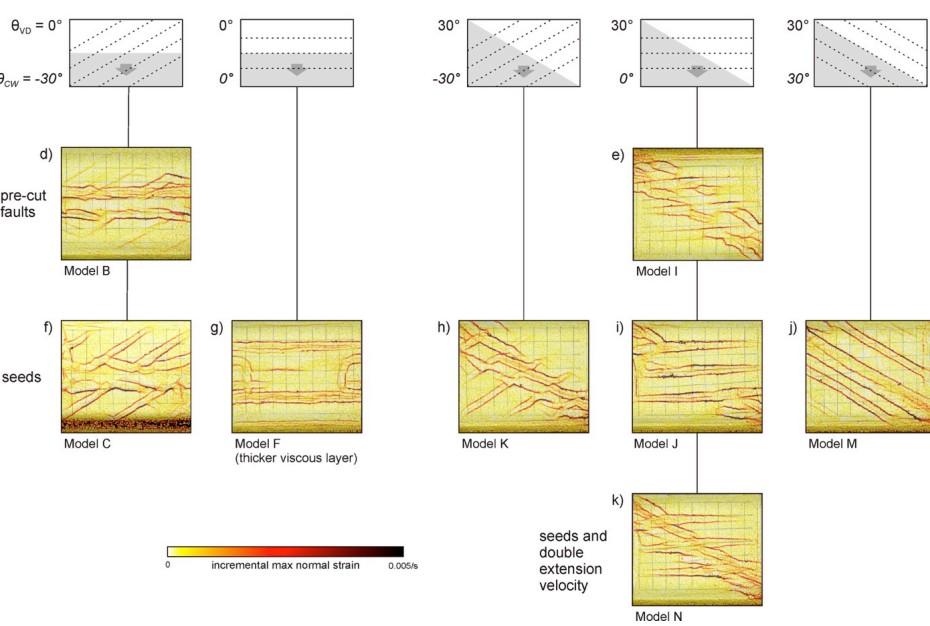

**Fig. 11. Summary of model results (PIV-derived incremental maximum normal strain) as a function of mantle discontinuity geometry, as well as type and geometry of crustal weakness, and extension velocity. Top views represent the incremental deformation during the final model stage (e = 10%), or during early model stage (e = 1.6%) in panels (a-c). Angle $\theta_{VD}$ provides the orientation of the VD (velocity discontinuity, representing a mantle weakness), whereas angle $\theta_{CW}$ provides the angle of the simulated crustal weakness (see Fig. 1c-f for definitions).**





Both the VD and the simulated crustal weaknesses localized deformation from the start of the model runs, but in different
        ways. Simulated crustal weaknesses usually dominate in the early stages of extension, while VD deformation trends appear
        later at the surface, once the strain from the VD has been transferred through the viscous layer into the sand cover. The
        resulting deformation is first diffuse but may later evolved into well-marked (offset) fault zones and graben (Figs. 5-10, 11d-
        j, 12a). By contrast, the pre-cuts and seeds directly induced localized deformation in the shape of faulting (Figs. 5-10, 11d-j).

        Furthermore, faster extension enhanced the influence of the VD on rift structure development (Fig. 11k). Increasing the
        extension velocity caused enhanced coupling between the base plate and the overlying materials. Hence the VD can
        overprint the otherwise dominant control of the seeds in these models (compare Fig. 11i and Fig. 11k). Conversely, slower
        extension rates should have the opposite effect, increasing the influence of weaknesses in the sand layer on the development
of rift structures.

### 4.2. Comparison with previous analogue and numerical studies

        The results from our models with only a VD are very much in line with previous analogue and numerical modelling work.
The double graben systems in our orthogonal extension models (Fig. 11a, b) reflect the double graben observed by e.g. Brun
        and Tron 1993; Michon and Merle (2000; 2003), Dyksterhuis et al. (2007) and Zwaan et al. (2019) (Fig. 12a, b). These
        double graben structures are due to the development of shear zones within the viscous layer that dip away from the VD
        (Michon and Merle 2003; Zwaan et al. 2019). Where these shear zones reach the brittle cover layer, they cause the
        development of a graben. This viscous shear zone also explains the wider spacing between both graben in the model with a
thicker viscous layer; assuming a constant dip angle, the shear zone transfers deformation farther away from the VD (Figs.
        11a, b, 12a, b). This viscous shear zone effect also occurs when the VD orientation is oblique to the extension direction, but
        here the early deformation zones are overprinted by en-echelon faults that are sub-perpendicular to the extension direction
        (Fig. 11c). Such en-echelon fault structures are typical for oblique extension systems and are reported in numerous analogue
        modelling studies (with or without base plate set-ups) and numerical work (e.g. Tron and Brun 1991; McClay and White
1995; Bonini et al. 1997; Clifton et al. 2000; Van Wijk 2005; Agostini et al. 2009; Autin et al. 2010; Corti et al. 2013; Brune
        and Autin 2013; Phillippon and Corti 2016; Duclaux et al. 2020). These studies also registered the general (counter-
        clockwise) rotation observed in our oblique VD models. Furthermore, the final en echelon double rift structure that
        developed in our oblique VD models (Fig. 11c) is found in analogue models with oblique mantle weaknesses by Autin et al.
        (2013) as well.





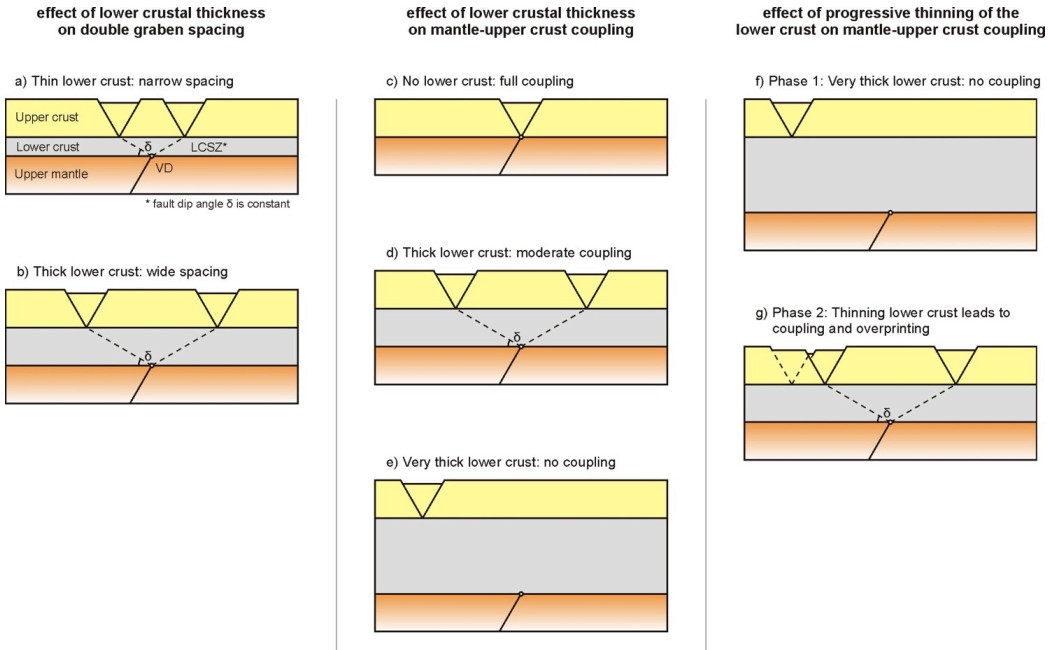

**Fig. 12. Schematic 2D sketches depicting how lower crustal thickness variations may affect rift structures. (a-b) A thicker lower crust widens the spacing between double graben related to lower crustal shear zones (LCSZ) originating from a mantle weakness simulated by a velocity discontinuity (VD) in our models. (c-e) The degree of coupling between the mantle and crust is strongly affected by lower crustal thickness: the thicker the lower crust, the less coupling occurs. When no coupling occurs, deformation in the upper crust can localize along crustal weaknesses only (e.g. Zwaan et al. 2019). Note however that coupling is also affected by extension velocity (e.g. Brun 1999; Zwaan et al. 2019, see Models F, G, N, O, Figs. 5a-j, 9, 11i, k). (f-g) Progressive thinning of the lower crust during rifting may cause a previously decoupled system to become coupled, so that previous upper crustal weakness-controlled structures may be overprinted by mantle-weakness controlled structures.**

The varying impact of the different types of crustal weaknesses on rift structures concur with previous models.

For example, Wang et al. (in review) show how the amount of extension in a previous deformation phase determines the magnitude of the inherited weaknesses. In their analogue models, smaller amounts of previous extension (e = ca. 4%) leads to less defined structures that have less effect on deformation during a subsequent deformation phase, whereas greater initial extension (e = ca. 12%) leads to more developed structures that are more likely to be reactivated and control subsequent faulting. The pre-cut faults in our models would represent the first case with less developed inherited structures only mildly affecting subsequent faulting (Fig. 11d, e), whereas the seeds in our models have much more impact and would represent the second option involving more developed structural inheritance (Fig. 11f-j). A similar effect of different grades of weakness seems to occur in the numerical rifting models by Dyksterhuis et al. (2007), although these models are more focused on the effects of large-scale weaknesses on general rift style (wide versus narrow rifting, see also Brun 1999).



The decreasing influence of a crustal or mantle weakness as it is oriented more obliquely to the extension direction is also observed in previous models. For instance, the studies by McClay and White (1995), Bellahsen and Daniel (2005), Henza et al. (2011), Zwaan and Schreurs (2017), Deng et al. (2018) and Molnar et al. (2019) show that a linear weakness that is (sub-

)parallel to the extension direction is unlikely to be reactivated, and if such a structure is reactivated it takes up less deformation than a structure oriented more oblique to the extension direction. Note however, that a VD can always force (some) deformation to occur in the overlying model materials (e.g. Acocella et al. 1999; Dauteuil et al. 2002). However, when both types of weaknesses are parallel, even when orientated obliquely to the extension direction, enhance the localization of deformation along both weaknesses (Figs. 11j). It is furthermore important to stress that in models with

oblique weaknesses only, these weaknesses account for all deformation, and no structures orthogonal to the regional direction of extension occur (Fig. 11h, j). Although these oblique structures resemble standard graben bounded by normal faults at first sight, yet they underwent some oblique slip. As far as we are aware, these complex features resulting from interacting oblique weaknesses have not been described in previous (modelling) studies.

However, the respective roles of mantle weaknesses, represented by the VD, and simulated crustal weaknesses in our Models C and D are in line with the observations from the analogue modelling work by Molnar et al. (2020), but we explore a much broader parameter space. Molnar et al. (2020) show that a broad viscous weak zone simulating a thermal mantle anomaly (having a similar function as the VD in our models, i.e. mimicking a mantle weakness), has a dominant effect during orthogonal rifting, whereas 30° obliquely oriented crustal weaknesses (seeds) have a secondary effect. This is identical to our

observations in Models C and D with a very similar set-up (Figs. 2k-t, 3, 11f). However we went a step further by testing various other weakness orientations and find that the simulated mantle weakness may not always affect surface structures during the early phases rifting, likely due to decoupling, a factor that is also present in the 2D numerical models described by Chenin and Beaumont (2013).

Our models also show that the extension velocity defines the degree of coupling between the mantle (VD) and the overlying materials. When extension is fast, the rheology of the viscous layer above the VD becomes more rigid (e.g. Brun 1999; Zwaan et al. 2019). As a result of this enhanced rigidity, the viscous layer no longer acts as a decoupling layer as it would if extension were slower. Now the viscous layer can directly transfer deformation into the sand layer, strongly localizing deformation along the VD and over-ruling the otherwise dominance of the seeds (Fig. 11k). Conversely, a decreased

extension velocity would decouple overlying layers from the VD, allowing more influence by the seeds (e.g. Zwaan et al. 2019, Fig. 12e). This decoupling may be similar to the tectonic situation simulated numerically by Liao and Gerya (2015).

Finally the results of our models that include differently oriented simulated mantle and crustal weaknesses bear resemblance to previous analogue models involving multi-phase extension and different extension directions (Fig. 11d-k), but this





resemblance does not stand scrutiny upon closer inspection. Although these previous studies produce different rift
orientations (e.g. Bonini et al. 1997; Withjack et al. 2017; Ghosh et al. 2020; Wang et al. *in review*), a very important
contrast is that these differently oriented rifts or normal faults were formed and active at different times in the evolution of
the model, whereas the differently oriented rifts in our models were generally active during most of the model evolution
(Figs. 2-10).


### 4.3. Model strengths and limitations

Although our models provide important insights into the influence of, and interaction between a mantle and crustal
weaknesses, there are inherent limitations. For example, we did not include syn-rift sedimentation. However, this is not
expected to have had a significant influence on early rifting since accommodation is limited (Zwaan et al. 2018a). Similarly,
no thermal effects, magmatism or isostatic effects of the mantle were simulated, which is permissible, as these are not
considered to be important during the early evolution of a rift zone. The same is true for the constant extension direction and
extension velocity in our models, which may vary during the long-term evolution of a rift system (e.g. Brune et al. 2016), but
can be considered constant during early rifting.

Our models cannot go beyond the initial stage of continental rifting; no thermal effects, phase changes, continental break-up
and oceanic spreading can be simulated. Although some analogue modellers have successfully simulated more advanced rift
stages (e.g. Brun and Beslier 1996; Molnar et al. 2017; Khalil et al, 2020), numerical methods are currently more suitable for
such late-stage rift modelling (e.g. Brune et al. 2016; Liao and Gerya 2015; Tetreault and Buiter 2018; Wenker and
Beaumont 2018; Chenin et al. 2019). Directly comparing our simple analogue models with these more complex numerical
models is therefore challenging.

However the relative simplicity of our model set-up has the benefit of allowing us to clearly distinguish the effects of
specific parameters on the subsequent rift structures. Furthermore, our results provide detailed information on internal and
external deformation in our models through the unique application of CT imagery and high-resolution PIV analysis. This
combination of techniques allows for an unprecedented detailed analogue model analysis, revealing the complexity of model
evolution that is not readily captured by directly observing top view images.




### 4.4. Implications for interpreting natural rift systems

Our analogue model results may have important implications for the interpretation of natural rift systems in which inherited structural weaknesses are present. In particular, the results show that differently oriented crustal and mantle weaknesses can
simultaneously localize deformation to create complex rift structures with different orientations (Fig. 11). At first sight, the complex rift structures observed at the final stage (similar to current-day field observations, see right-hand columns in Figs. 2, 4, 5, 7, 9) would suggest a multiphase rifting history with a changing extension direction to explain the different fault sets orientations, in line with analogue models (e.g. Bonini et al. 1997; Withjack et al. 2017; Ghosh et al. 2020; Wang et al. *in review*). Such changing extension directions have been invoked to explain the current state of various similarly complex
natural examples (e.g. Roberts et al. 1990; Bonini et al. 1997; Erratt et al. 1999; 2010, and references therein). Yet our models illustrate that such complex rift structures can also form during a single continuous rift phase with a constant extension direction. This means that a multiphase rift evolution involving different extension directions is not always required to explain different structural orientations in rift basins.

Moreover, our model results indicate that the type of weakness, as well as its orientation with respect to the regional extension direction is important. Studies of natural examples confirm that not all inherited weaknesses present in a system need to reactivate (Wilson 1966; Bell et al. 2014; Wang et al. in review), and the observation that normal faults do not require to be (sub-)perpendicular to the regional extension direction (Fig. 11h, j) shows that structural orientations themselves might not always be reliable indicators for past extension directions. These important insights derived from our
models suggest that we may need to reassess the tectonic history of various complex rift structures in nature. In particular, attention should be dedicated to detailed fault activity analysis such as recently performed by e.g. Claringbould et al. (2017, 2020) and Phillips et al. (2019) in order to distinguish whether important changes in extension direction did occur.

A further model observation useful for the interpretation of natural rifts are the effects of coupling between the simulated
mantle and crust. As shown in our model, such coupling is a function of extension velocity (Figs. 4a-e, 9, 10, 11i, k), but also depends on the thickness and viscosity of the ductile lower crust (e.g. Brun 1999; Zwaan et al. 2019, Fig. 12c-e). It follows that decoupling between the mantle and crustal layers in natural settings may cause the effect of mantle weaknesses to be less apparent during the early stages of extension (Fig. 12e). Hence crustal weaknesses can dominate the rift structure arrangement at the surface during early rift stages. Conversely, strong coupling between the mantle and crustal layers may
overrule the effect of any crustal weaknesses if a weak heterogeneity exists in the stronger upper mantle (Chenin and Beamont 2013; Zwaan et al. 2019, Fig. 12c). We would like to highlight that also these small but significant details would be hard to distinguish when interpreting the final model stage, or a natural example. A further important possibility is that the general thinning of the lower crust during initial rifting (or changes in extension velocity, e.g. Brune et al. 2018) may cause





increased coupling over time (Sutra et al. 2013), so that initially dominant crustal weaknesses may be overprinted by mantle
influences as rifting progresses (Fig. 12f, g).

## 5. Conclusion

In this paper we presented a series of brittle-viscous analogue models to study how differently oriented mantle and crustal
weaknesses may interact and affect rift development. Our model results bring us to the following conclusions:

- In the absence of simulated crustal weaknesses, the modeled rift system is governed by the velocity discontinuity or VD representing a mantle weakness. The resulting structure is a rift zone parallel to the VD. In case of VD orthogonal to the extension direction, this rift zone consists of large, normal faults parallel to the extension direction. In contrast, an oblique VD leads to en-echelon normal faulting at an angle to both the VD and the
extension direction, but that tends to align perpendicular to the extension direction (counterclockwise rotation).
- Simulated crustal weaknesses can (partially) overprint the general rift structure developing along the VD. The larger the decrease in strength due these crustal weaknesses with respect to the undisturbed brittle layer, the larger their control on the resulting rift pattern. In general, crustal weaknesses localize faulting first, after which the VD-induced fault develop and interfere.
- The orientation of the simulated mantle or crustal weaknesses with respect to the extension direction has a significant impact on its reactivation. A weakness that is oriented orthogonally to the extension direction is likely to localize normal faulting, whereas an oblique weakness is less likely to do so. Yet if mantle and crustal weaknesses are parallel, they might amplify each other's effect and if both weaknesses are oblique, no faulting orthogonal to the extension direction may occur. As a result, structural orientations are not always indicative of past extension
directions.
- The coupling between the mantle and the overlying crustal layers determines the relative influence of the crustal and mantle weakness (VD) on rift evolution. When coupling is strong (e.g. due to fast extension), the VD is the dominant weakness, whereas crustal weaknesses become more impactful when coupling is low. Coupling may vary over time due to progressive thinning of the lower crustal layer, as well as due to variations in extension velocity, so
that early structures controlled by crustal weaknesses can be overprinted by later mantle-controlled structures.
- Most importantly, our models show that crustal and mantle weaknesses with different orientations can both simultaneously produce rift structures. This means that in order to explain different structural orientations in rift basins, multiphase extension is not always required. These findings, together with the observation that structural trends should not always reflect past extension directions, provide a strong incentive to reassess the tectonic history
of various natural examples.





**Data availability:**

Images and videos of the models, including PIV analysis results and CT-imagery, are freely available in the shape of a data publication stored by GFZ Data Services (Zwaan et al., 2021). Link: **DOI LINK TO BE CREATED**

**Author contribution**

FZ: Conceptualization, Formal analysis, Investigation, Methodology, Validation, Visualization, Writing – original draft preparation

PC: Conceptualization, Methodology, Writing – review & Editing

DE: Conceptualization, Methodology, Writing – review & Editing

GM: Conceptualization, Methodology, Writing – review & Editing

GS: Conceptualization, Methodology, Funding acquisition, Resources, Supervision, Writing – review & Editing

**Competing interests**

The authors declare that they have no conflict of interest

**Acknowledgements**

We thank Nicole Schwendener for her assistance during the CT-scanning process, and to Timothy Schmid, Michael Rudolf, Matthias Rosenau, as well as the software engineers from LaVision (Dave Hollis, Horst Nagel, Torsten Siebert) for technical support during the PIV analysis in DaVis. We are also grateful to Kirsten Elger for her help in creating the GFZ data publication containing the supplementary material (Zwaan et al. 2021). This research was funded by the Swiss National Science Foundation (grant 178731, http://p3.snf.ch/Project-178731), which also covered the Open Access publication costs.




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
