# Peer review of "Complex rift patterns, a result of interacting crustal and mantle weaknesses, or multiphase rifting? Insights from analogue models"

_Solid Earth, 2020_

## Author Comment (AC1)

**Subject: Comment on se-2020-214**

Review of "Complex rift patterns, a result of interacting crustal and mantle weaknesses, or multiphase rifting? Insights from analogue models"

**Summary:**
This manuscript addresses the problem of understanding the effects of inherited weaknesses on rift evolution. The authors produce 3D analogue models to test the interaction between differently oriented in the crust and upper mantle. The main result is that crustal and mantle weaknesses can simultaneously localize rift structures leading to intricate fault patterns that could be interpreted as a result of multiphase extension. The authors conclude that multiphase extension is not required to explain different structural orientations in rift basins, and suggest that the tectonic history of natural examples should be reevaluated.

**General Comments:**
This is an interesting manuscript that aims to solve an important problem in the evolution of rift basins. I think that the manuscript is well-written, explains clearly the methods used, and arrives at reasonable conclusions. The implications are well-received and useful for interpreting deformation patterns in rift basins.

- **REPLY:** We thanks reviewer 1 for taking the time to go trough out work, and for submitting this positive review.

One component that I think is missing is the comparison and contrast of the resulting models with natural rift patterns. For example, the comparison with the Malawi Rift would be interesting because it is a young rift with low extension (just as the set-up of the models) and inherited crustal weaknesses with varying orientations. The trend of the Malawi Rift is perpendicular to the extension direction but it meets the Shire Rift to the South with an oblique orientation. Do the rift structures show a pattern recognizable in the analogue models in this manuscript? The addition of this component would increase the impact of the manuscript.

> **REPLY:** We understand the request to expand the comparison between our model results and nature. However, the goal of this work is not to directly compare our model results with nature. Instead, the analogue models presented in this manuscript aim to explore the general impacts of these parameters (hence we also included a comparison with previous analogue and numerical modelling efforts). We hope that our model results will inspire our colleagues to revaluate the tectonic history of various natural rift systems (and rifted margins). We believe however, that adding a detailed comparison with a number of natural examples would distract from the main, more general, results and would significantly lengthen the text, which we deliberately kept as to-the-point as possible. Adding such model-nature comparisons would also be challenging, as the presence and character of crustal and mantle weaknesses are often not very well known in nature.

- However, since also reviewer 2 has requested the inclusion of more natural examples, we decided to mention a number of natural examples that could be useful for further studies and comparisons. In future work, we plan to go one step further and compare our results with natural examples, however, this will require another approach and a careful analysis of the natural examples which is, as said before, not possible in the frame of this contribution

**Grammatical and orthographical errors on lines.**

Paragraph 500: Fix ")parallel"

- **REPLY:** Thanks for noticing, the word "(sub-)parallel" is spread over two lines, and word automatically cuts it this way. We will double-check the final paper.

---

## Author Comment (AC2)

**Subject: Comment on se-2020-214**
**Preview your Referee comment**

Review – Zwaan et al – Complex rift patterns, a result of interacting crustal and mantle weaknesses, or multiphase rifting? Insights from analogue modelling

This study uses analogue modelling to investigate the relative, and often competing effects of crustal and mantle weaknesses on rift physiography. The paper addresses an important question in structural geology and tectonics, namely whether multiphase rifting is required to explain non-collinear faults and rift systems, and proposes that such systems may form due to the interplay of crustal and mantle weaknesses during a single phase of rifting.

The paper presents a detailed and comprehensive study of a series of models and thoroughly explores a range of parameters. This study will be of wide interest to people interested in structural geology and tectonics and the model observations will have implications for a wide range of rift systems. I list a few general comments and suggestions on the manuscript below, before giving some more detailed line-by-line comments.

If the authors have any questions or if anything is unclear, feel free to get in touch.
I believe that the manuscript represents a detailed and thorough piece of work and will be of interest to the Solid Earth readership. On that basis I suggest the paper should be accepted for publication after consideration of the following comments:

**General comments – these are expanded upon further in the individual comments.**

The interactions between the weaknesses and faults should be expanded to discuss more aspects of how the weaknesses/seeds may not reactivate but still segment and block the propagation of faults and rift segments, i.e. they are not reactivated but are also not entirely passive and cross-cut during the rifting. This could be expanded in either the introduction or discussion, but these high-angle structures represent important weaknesses throughout breakup in that they may influence the site of future transfer zones.

- **REPLY:** We have added a couple of remarks concerning segmentation, but it is challenging to establish a proper comparison with natural examples, since most field studies focus on crustal structures in the "basement" (i.e. the metamorphic/crystalline upper crust), whereas we study the interaction between crustal and mantle weaknesses. The latter are often poorly defined, not only because there is a lack of data but also because it is difficult to define a mantle "inheritance" in a natural system. Segmentation is also often reported in multiphase extension models that have no clear mantle weakness (but rather a distributed extension basal boundary condition). We prefer to not go into too much detail here, since the models are meant to provide a general overview of the influences of the the various model parameters, and small details may not be significant or robust, but due to randomness and chance that affect all modelling studies.
- It is true that not all weaknesses do (fully) reactivate, but this is rather rare. In most models they do and we do describe how the orientation of the weaknesses is a very important factor controlling reactivation.
- Concerning the transfer zones: we are not sure if the reviewer refers to transfer zones between individual rift segments/basins, or the development of oceanic transform zones (as he refers to break-up). We mentioned "transform zones" in the introduction (related to the preceding work by Molnar et al. 2020), and we made sure to mention it once more in the discussion. We have also added some more attention to the segmentation of the rift basins.

The authors provide a detailed and comprehensive comparison to previous modelling studies. However, I think the study would benefit from an increased comparison to rift basins and natural examples, particularly in the discussion. There are a number of parallels here with rift systems in

East Africa and the North Sea, amongst other areas. More direct comparisons should be drawn between specific natural examples and the model results.

- **REPLY:** We understand the request to expand the comparison between our model results and nature. However, the goal of this work is not to directly compare our model results with nature since this is not a simple task, since the reviewers ask to analyse and define the natural system to an extent that is not possible in the framework of this contribution. Such an undertaking would represent a paper by itself. The aim of this study is to present the analogue models to explore the general impacts of these parameters (as such we did include a comparison with previous analogue and numerical modelling efforts).
- We hope that our model results will inspire our colleagues to revaluate the tectonic history of various natural rift systems (and rifted margins). We believe however, that adding a detailed comparison with a number of natural examples would distract from the main, more general, results and would significantly lengthen the text, which we deliberately kept as to-the-point as possible. Adding such model-nature comparisons would also be challenging, as the presence and character of crustal and mantle weaknesses are often not very well known in nature.
- However, since also reviewer 1 has requested the inclusion of more natural examples, we decided to mention a number of natural examples that could be useful for further studies and comparisons.

Heron et al., (2019) use numerical modelling to discuss the relative importance of crustal and mantle weaknesses in the evolution of the Labrador Sea. It would be interesting to see how your observations from an analogue modelling perspective compare to those generated in numerical models.

- **REPLY:** Many thanks for suggesting this paper. Although Heron et al. (2019) test a very different set-up (specifically reproducing the Labrador Sea setting with two extension phases that are 90° oblique), there are some interesting similarities (e.g. weaknesses oriented more parallel to the extension direction tend to be less activated, and if crustal and mantle weaknesses are oriented parallel, they enhance each other). We have included various references in the revised manuscript.
-

**Line by line comments**

L25 – what do you mean by great depth and high temperature here, can you give examples and be more specific?

- **REPLY:** We modified the text: These initial weaknesses may be situated anywhere in the lithosphere, although structural heterogeneities tend to be attenuated or erased at great depth where temperature is high (T > 800 +/- 300 °C depending on the geothermal gradient, the nature of the rock involved and the extension rate; Braun et al 1999; Yamakasi et al. 2006).

Paragraph 2 – Worth mentioning the work of Heron et al, looking at the influence of "perennial mantle scars"

- **REPLY:** Thanks for the suggestion, we have added references to Heron et al. (2016) and Heron et al. (2019), and the many references therein.

L40 – Schiffer et al., (2020) may be relevant here, looking at structural inheritance (including crustal and mantle structures) in the North Atlantic.

- **REPLY:** We have included a reference to the interesting review paper in the revised manuscript.

Para 2, Line 43 – Can you expand on why crustal structures may not reactivate? – Can these structures still influence the rift, i.e. through segmentation/blocking?

- **REPLY:** The reasons for not reactivating would be the same as for mantle weaknesses: if not properly oriented, or not affecting (i.e. weakening) the strength of the crust sufficiently. This was already implied in the text, but we have rephrased things to avoid confusion.

L72 – Throughout the introduction, more emphasis should be given on multiphase rifting as a concept and the ways in which non-collinear fault systems may form through this process, as well as crustal/mantle weaknesses. Reeve et al. (2015) showcase a number of mechanisms that may contribute to non-collinear fault systems, similar to those produced in the models.

- **REPLY:** We agree that the concept of multiphase rifting should be highlighted a bit more in the introduction. Many thanks for pointing us to this very interesting paper by Reeve et al. (2015). We have added a paragraph on interpreting rift basins and how this may be challenging due to the influence of structural weaknesses on the one hand, and changing extension directions on the other hand.
- However, we aim to keep our study and therefore also the manuscript at this state more focused to the model results and not to go into too much detail in order to not distract and defocus the reader away from the main aim of the paper.

Section 2.2 – Can you expand on what the velocity discontinuity is equivalent to in reality? Could this be equated to a thickness change in the lithosphere, akin to the Sorgenfrei-tornquist Zone?

- **REPLY:** The VD used in analogue models is generally considered to be equivalent to a large-scale shear zone or fault zone in the strong upper mantle, as we now describe in the text. This VD could be an old suture that is reactivated, but perhaps also a new structure that develops in a zone where the lithosphere (that is, the upper mantle) is thinner and thus weaker. Alternatively, it could also a change in mantle rheology/strength due to difference in upper mantle composition. The VD has often been used by modellers, but is to a degree a "trick" to force localization, akin to the "seeds" or pre-cuts that we use to simulate crustal weaknesses, or the "seeds" or pre-existing weaknesses applied in numerical models.

Very detailed explanation of the model setup, and comprehensive descriptions of the scaling properties of the model.

- **REPLY:** Thanks for the positive feedback

**Results section** – Would be useful to the reader to briefly outline the different models and the key things that they examine at the start of this section, before delving into the results.

- **REPLY:** We have now added a short description at the start of this section.

Figure 2 – would be useful to distinguish the pre-cut faults from the weak crustal seeds on the figure. One suggestion would be to change the markers along the sides of the model to allow the two to be distinguished easily.

- **REPLY:** We have now changed the markers, and added annotation (See also reply to next comment)

Line 233 – What is the difference between Models C and D in terms of setup? Are they the same model but ran twice? Would be useful to explain why this is the case. Ah, I see this is related to the CT scanning. Can you make this clearer on the figures to ensure that people are not looking for differences that are not there (Also for Models G and H). Figures would also benefit from a cross-section of the initial setup for each of the models. This is particularly the case for Figure 4 which has a thicker lower crust.

- **REPLY:** We have specified that Model D is a rerun of Model C in the text, and we did the same for models G and H, and added this information to all captions of the CT-scan figures. We have made this clearer in the figures.
- In the result images, it would perhaps be optimal to add an extra column to the left describing the set-up, but this would reduce the size of the other figure panels to such a degree that things would become too small, therefore we did not include it.
- Cross-sections could potentially be of help, but are not always helpful when presenting a 3D model set-up.
- We have instead opted to add additional annotations to the figures and made things as clear as possible in the captions, but we also kept this to a minimum to avoid crowding the images too much.

Good exploration of the available parameter space.

- **REPLY:** Many thanks for the positive feedback

Model J – Do you have any information on the displacement along the faults in the model? It would be very interesting to see whether there is a change in displacement as the faults cross the VD.

- **REPLY:** When looking closely at the PIV images, it seems that displacement (as defined by the maximum incremental normal strain), are often slightly decreased where the seed crosses the VD.However, we think describing and discussing this would distract from the general description, but it will be a point of interest for future work.

Line 340, figure 7 – Model K shows some fascinating features and very clear evidence of rift segmentation. Would be interesting to see how this compares to observations of rifts where crustal weaknesses have been proposed to segment rifts (i.e. the Viking Graben in the North Sea, Phillips et al., (2019), Fossen et al., 2016)

- **REPLY:** We have expanded the discussion a bit to better address this (section 4.1, 4.2 and 4.4.), but prefer to avoid going into too much detail (See general reply).

Model N – Initial seed-related faults are partitioned by the VD. Is there a switch in polarity occurring across this discontinuity? Would you be able to expand on this slightly?

- **REPLY:**
- There may indeed be a slight shift in strain values when the seed-related faults cross the VD. This could be due to an interaction with the VD-related faults.
- However, we think describing and discussing this in detail would distract from the general description, but it will be a point of interest for future work.

**Discussion**

Line 416 – I assume that the right-stepping nature of the en-echelon faults is a symptom of the orientation of the weaknesses rather than a fundamental feature? This should be made clearer.

- **REPLY:** The right-stepping en echelon faults are indeed a result of the oblique orientation of the VD (simulated mantle weakness). If the orientation of the VD would be mirrored (angle of -30˚, rather than 30˚, see set-up in Fig. 1), the en echelon orientation would be left-stepping. We now mention the 30˚ orientation in section 4.1, and added some text to section 4.2 to clarify this.

Figure 11 – Nice, clear summary diagram. Might be worth annotating/labelling key features and observations onto the diagram to make it clearer.

- **REPLY:** We have added some annotations to the figure as suggested.

Line 463 – Can you be more specific here, it seems to me that they appear to dip downwards towards the VD. Are you referring to the structures from the bottom up?

- **REPLY:** We are indeed referring to the structures from the bottom up, i.e. the shear zones starting from the VD, reaching to the base of the brittle layer while crossing the viscous lower crustal layer. This is now specified in the text.

Line 490 – Would be interesting to see how this compares to the crustal weaknesses as modelling in Henza et al, 2010, 2011 too

- **REPLY:** The model results from Henza et al. (2011) indeed show how less developed structural weaknesses from an initial deformation phase have less impact on subsequent deformation. We have included this in the discussion. However, Henza et al. 2010 use constant amounts of extension in both phases, hence we decided to not cite it here.

Line 502 – check sentence structure here, seems like something is missing.

- **REPLY:** We believe the original sentence was correct, but have modified it slightly to avoid confusion.

Line 509 – Would be good to mention the segmentation aspect of the higher-angle weaknesses at some point here.

- **REPLY:** We added a mention of segmentation to the subsequent paragraph (the comparison with Molnar et al. 2020)

Line 568 - See Reeve et al., (2015) for potential mechanisms that may give rise to non-collinear fault systems.

- **REPLY:** We have included a reference to Reeve et al. (2015) in order to closely tie the ideas presented in this previous paper to our revised manuscript.

**References**

Heron et al., (2019) -  https://doi.org/10.1029/2019TC005578
Schiffer et al., (2020) - https://doi.org/10.1016/j.earscirev.2019.102975
Reeve et al., (2015) - https://doi.org/10.1016/j.jsg.2014.11.007
Phillips et al., (2019) - https://doi.org/10.1029/2019TC005756

Fossen et al., (2016) - https://doi.org/10.1144/SP439.6
Henza et al., (2010) - 10.1016/j.jsg.2009.07.007
Henza et al., (2011) - 10.1016/j.jsg.2011.06.010